# Barriers and facilitators of primary care management of type II diabetes mellitus in the West African sub-region: A scoping review

**Abdul-Basit Abdul-Samed**[1]*, **Yasmin Jahan**[2], **Veronika Reichenberger**[2],
**Ellen Barnie Peprah**[1], **Mary Pomaa Agyekum**[1], **Henry Lawson**[1], **Dina Balabanova**[2],
**Tolib Mirzoev**[2], **Irene Akua Agyepong**[1]

**1** Ghana College of Physicians and Surgeons, Accra, Ghana, **2** London School of Hygiene & Tropical Medicine, London, United Kingdom

* abasit@gcps.edu.gh

## Abstract

The prevalence of diabetes is rising rapidly across West Africa, posing a significant public health challenge. Effective diabetes management through accessible and quality primary healthcare is crucial, yet multiple barriers persist. This review aimed to synthesise the available evidence on factors influencing access, utilisation, and quality of diabetes primary care in West Africa. Following Arksey and O'Malley's framework and PRISMA-ScR guidelines, we searched four electronic databases (PubMed, Scopus, Google Scholar, CAIRN Info) and grey literature sources. Eligibility criteria included: peer-reviewed studies published between 2000–2023 in English or French; primary research focusing on adult type II diabetes care in West African countries; and studies reporting on factors affecting access, utilisation, or quality of primary healthcare. Data were extracted using a standardised form and analysed through framework synthesis integrating the WHO Primary Health Care Framework, Social Determinants of Health model, and Innovative Care for Chronic Conditions model. Twelve studies were included from Nigeria (n=7), Ghana (n=4), and Senegal (n=1). Key barriers to access, utilisation, and quality were identified as health system deficiencies, including inadequate infrastructure, workforce shortages, supply gaps, fragmented coordination of care, absence of standardised guidelines, high costs of care, and inefficient leadership/governance for chronic disease management. Broader determinants of health, such as poverty, gender, cultural beliefs, reliance on traditional medicine, and health policy gaps, significantly influenced access to and utilisation of care. Individual-level barriers like psychological distress and delays in care-seeking were also significant. Family/social support systems emerged as potential facilitators of accessing and utilising PHC services. Our review identified that to improve diabetes care, West Africa needs context-specific models that align indigenous healing practices with PHC, strengthen health systems, and address sociocultural determinants. Future research should focus on developing and evaluating culturally resonant interventions that

**Data availability statement:** All relevant data are within the paper and its Supporting Information files.

**Funding:** This research was funded by the NIHR Global Health Research Centre for Non-Communicable Disease Control in West Africa using UK aid from the UK Government to support global health research. Funding was received from the National Institute for Health Research (NIHR) Global Health Research Centre (grant number NIHR203246 to TM) on Strengthening of Capacity for NCD Control in West Africa (Stop-NCD) (https://nihr.ac.uk/). The focus of this research is to strengthen the capacity for NCD control in West Africa. The views expressed in this publication are those of the author(s) and not necessarily those of the NIHR or the UK government. The funders had no role in study design, data collection and analysis, decision to publish, or preparation of the manuscript.

**Competing interests:** The authors have declared that no competing interests exist.

can navigate both biomedical and sociocultural factors shaping diabetes management in resource-constrained settings.

## Introduction

The global burden of diabetes has reached critical levels, with over half a billion adults living with the condition worldwide. This burden is particularly pronounced in low and middle-income countries (LMICs), including those in Sub-Saharan Africa (SSA) [1]. The World Health Organization (WHO) African region faces a significant double burden of disease, with communicable diseases remaining a major challenge while noncommunicable diseases (NCDs) rapidly increase in prevalence and impact. As of 2019, NCDs accounted for approximately 37% of all mortality in the region, up from 24% in 2000, with diabetes being a significant contributor alongside cardiovascular diseases, cancers, and chronic respiratory diseases [1,2].

In West Africa, precisely, the prevalence of undiagnosed diabetes is estimated at 4% and rising, with further estimates asserting that prevalence rates of undiagnosed diabetes in this region outpace rates in the rest of Africa [3–5]. A systematic analysis [3] found the pooled prevalence of undiagnosed diabetes mellitus across West Africa to be 4.72%, higher than Eastern Africa (4.43%), Northern Africa (4.27%), and Southern Africa (1.47%). Furthermore, prevalence varies across West African countries, with estimates ranging from 2.8% to 3.95% in Ghana, 1.7% to 2.1% in Burkina Faso, and as high as 5.6% in Niger [6–8].

Uncontrolled diabetes can lead to severe complications, including cardiovascular disease, kidney failure, blindness, and premature death [9,10]. These complications impact individual quality of life and substantially burden healthcare systems and societies. Effective primary health care (PHC) is crucial for preventing complications in ambulatory care-sensitive (ACS) conditions like type II diabetes.

The WHO recognises PHC as the most inclusive, equitable, and cost-effective approach to achieving universal health coverage (UHC). The WHO's PHC Operationalisation framework integrates three key components: integrated health services, multisectoral policy and action, and empowered communities. These connect to levers such as political commitment, governance, funding, and stakeholder involvement. This framework can enhance primary care's quality, equity, and effectiveness when functioning effectively. However, LMICs face significant challenges in implementation, particularly in managing chronic conditions like diabetes [11,12].

Compared to high-income countries (HICs), LMICs struggle with providing accessible, uninterrupted quality care for managing ACS conditions like diabetes that require lifelong self-care and continuous health system support [13]. West Africa, in particular, has considerable barriers hampering access to quality PHC, consequently creating substantial gaps in the delivery of effective first-line services for diabetes [14,15]. Health system barriers include limited resources, a shortage of human resources trained to manage diabetes, inadequate infrastructure, equipment, tools and supplies, and essential medicines [16–18].

Cultural factors also significantly influence engagement with health systems, sometimes orienting patients towards traditional healing rather than PHC services [19,20]; such beliefs and practices could contribute to delayed and interrupted care [19]. However, when appropriately leveraged, these cultural beliefs and practices may facilitate care-seeking through health system adaptation and community engagement [21,22].

Quality diabetes care requires addressing challenges at multiple levels - from health worker skills and medication availability at the health system level to social support systems at the community and individual levels [23]. A holistic view spanning micro to macro factors is necessary to ensure patient-centred care, as Hanefeld et al. argue - who emphasise that quality is co-produced by various actors and shaped by sociocultural contexts [24].

While studies have examined diabetes prevalence and risk factors in Africa, there is a notable lack of synthesised evidence on barriers and facilitators affecting diabetes-related primary care across West Africa [25]. This gap hinders the development of effective interventions and policies.

This review aims to inform targeted, context-appropriate strategies for diabetes prevention and care in West African PHC settings while contributing to the broader global understanding of managing chronic diseases in resource-constrained environments.

## Methodology

This scoping review followed the methodology outlined in our published protocol (S1 Protocol) [26], which adhered to the framework by Arksey and O'Malley [27], guidance from the Joanna Briggs Institute [28], and the PRISMA-ScR guidelines (S1 Checklist) [29].

### Stage 1: Identifying the research questions

Our review sought to answer two key questions:

1. What factors influence primary healthcare access, utilisation and quality for diabetes in West Africa?

2. How and why do these factors work, and what are the gaps in the current literature?

These questions were developed through an iterative process to comprehensively synthesise barriers and facilitators and help inform strategies to improve PHC.

### Stage 2: Identifying relevant studies

We conducted comprehensive searches across multiple electronic databases including PubMed, Scopus, Google Scholar, and CAIRN Info. The search strategy, developed with the assistance of an experienced librarian, was adapted for each database and utilised a combination of keywords and controlled vocabulary terms. The complete search strategy for PubMed, including all search terms, Boolean operators, and limits used, is provided in the Appendix (S1 Appendix). We also searched grey literature sources and reference lists of included studies and relevant systematic reviews.

### Stage 3: Study selection

The inclusion and exclusion criteria are detailed in Table 1. The review focused on original peer-reviewed primary research published in English or French between January 1, 2000, and December 31, 2023. We included qualitative, quantitative, and mixed-methods studies examining factors influencing diabetes care in the adult population, excluding gestational diabetes. Secondary research such as systematic reviews and meta-analyses were excluded to avoid duplicate reporting of findings.

The selection process involved three stages: Deduplication, Title/Abstract Screening, and Full-Text Screening. A team of six members worked in pairs to conduct the screening, with conflicts resolved through discussion and consensus with a third reviewer. The search results were exported into Rayyan software for screening and management [30]. A PRISMA flow diagram illustrates the study selection process (Fig 1).

### Stage 4: Charting the data

Data extraction was performed using a standardised form in Microsoft Excel. The form captured:

- Study characteristics (author, year, country, design, population)

- Barriers and facilitators to diabetes care access

**Table 1 . Inclusion & exclusion criteria.**

|  | Inclusion | Exclusion |
|---|---|---|
| Year of publication | 2000 - 2023 | N/A |
| Language | English, French | Other Languages |
| Population | Adults with diabetes | People with gestational diabetes and children |
| Concept | Access<br>Utilisation<br>Quality<br>PHC | N/A |
| Context | West Africa | Regions outside West Africa |
| Study design | Primary research studies:<br>• Qualitative studies<br>• Quantitative studies<br>• Mixed-methods studies<br>• Social science case studies<br>• Multi-country studies that include West African countries | • Studies that do not contain empirical data<br>• Systematic reviews and meta-analyses<br>• Scoping reviews and protocols<br>• Literature reviews<br>• Abstracts<br>• Commentaries<br>• Opinion pieces |

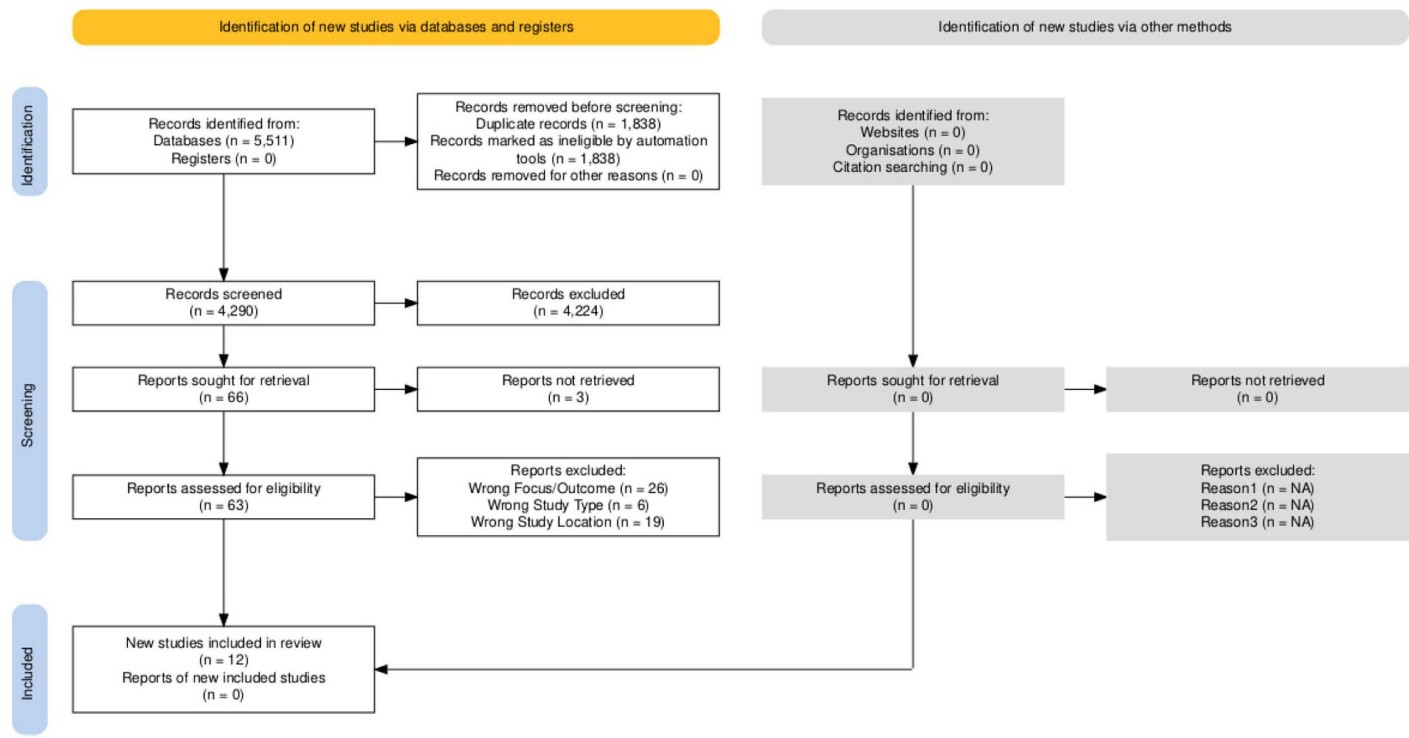

**Fig 1 . PRISMA flow diagram.**

- Utilisation and quality factors at health system and contextual levels

- Key findings and conclusions

   Four reviewers independently extracted data, which was then collated and summarised in a table. Further details of our methodology are available in our published protocol [26].

### Stage 5: Collating, summarizing, and reporting results

We utilised a framework analysis approach to direct the analysis process for this scoping review. To gain a holistic and context-specific understanding, various factors were analysed at the macro (broader societal, economic, political, cultural, and environmental factors influencing health and healthcare), meso (organisational and community level factors), and micro (individual patient, family, and healthcare provider level factors) levels. This taxonomy drew on multiple complementary frameworks: the Operational Framework for PHC, which is an expansion of the WHO's six building blocks of health systems [11], the Social Determinants of Health (SDH) framework [31], and the Innovative Care for Chronic Conditions model (ICCC) [32].

These frameworks were selected because of their relevance to the complexity of diabetes care. We combined the three frameworks by examining their key domains and concepts, looking for similarities and differences. The WHO PHC framework served as the primary structure for health system components (11), while the SDH model allowed us to contextualise these within broader societal factors (32). The ICCC model offered specific insights into chronic disease management relevant to diabetes care (33).

We then created a unified scheme incorporating elements from all three frameworks. For example, the 'health workforce' from the WHO PHC framework was expanded to include sociocultural competencies highlighted in the SDH model and the concept of a 'prepared, proactive practice team' from the ICCC model.

During data extraction and analysis, we applied this integrated scheme to capture the multifaceted nature of factors influencing diabetes care. In our synthesis, we carefully considered how the findings were related to multiple framework components. For instance, a finding related to patient adherence is analysed in terms of health system factors (WHO PHC), social determinants (SDH), and self-management support (ICCC).

This synthesised approach helped us better understand the complex factors influencing diabetes care in West African primary healthcare settings.

Our analytical framework distinguishes two main categories: contextual and health system factors across the macro, meso, and micro levels (Fig 2). Context refers to the surroundings or milieu in which a particular event or phenomenon is situated and is a crucial factor in shaping health and healthcare [33,34].

## Results

Twelve papers were included in this scoping review after screening 4,290 documents through our systematic selection process. Most of the included studies (n=7) were from Nigeria, with the remaining studies conducted in Ghana (n=4) and Senegal (n=1). The study designs employed across the various studies included qualitative (n=7), quantitative (n=4) and a mixed-method study (n=1), with study populations predominantly involving healthcare providers and patients.

The findings of this scoping review are organised and presented according to the integrated conceptual framework. We explicitly linked each key finding to relevant framework domains to provide a comprehensive understanding of the factors influencing diabetes care in West African primary healthcare settings (Tables 2 and 3).

To provide a comprehensive overview of our findings and their relationship to the integrated conceptual framework, we have synthesised the key results in Table 3

### Health System Factors

Our review identified several barriers within the health system that contribute to suboptimal primary healthcare for diabetes in West Africa. As outlined below, these barriers

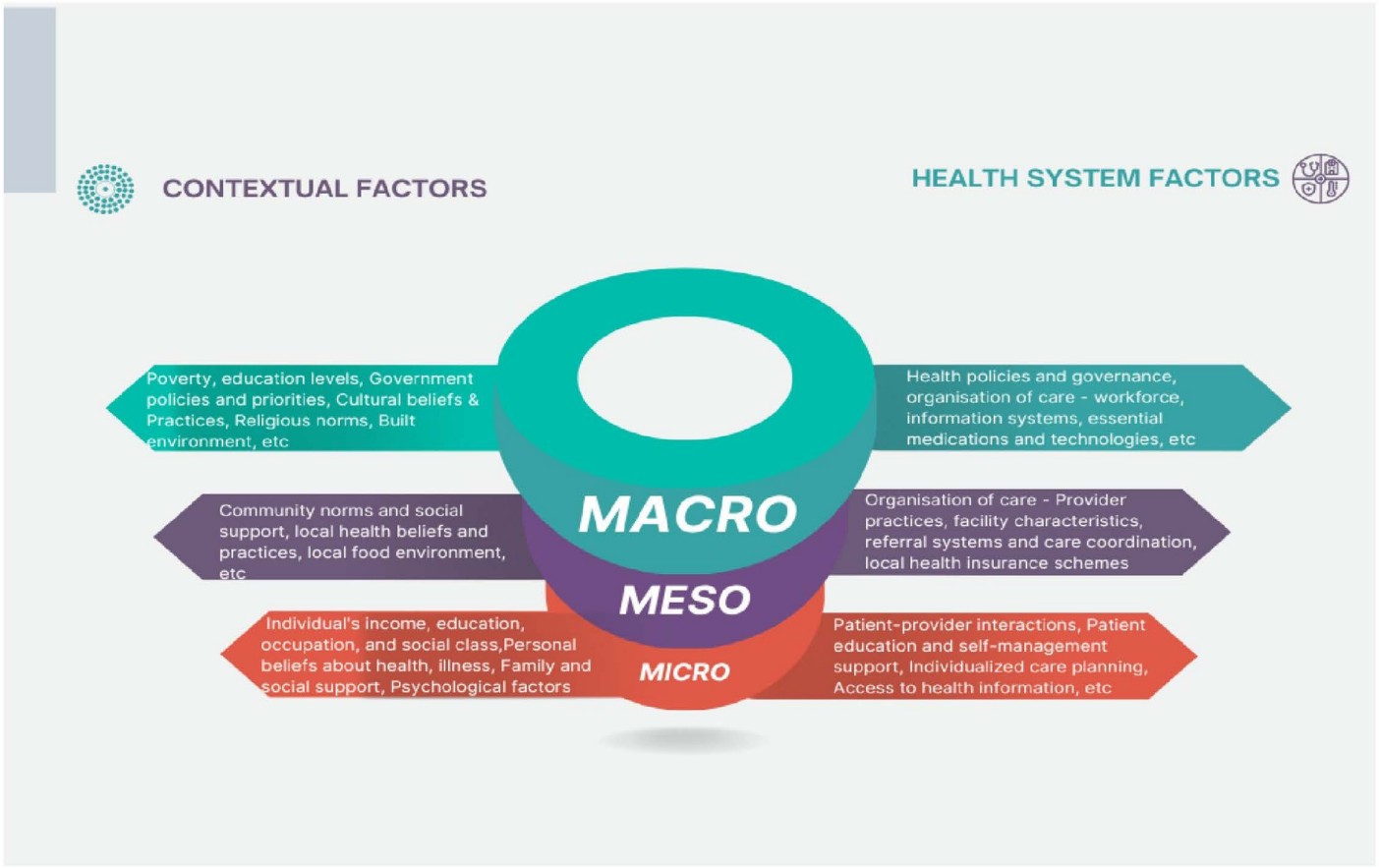

**Fig 2. Structure for analysis of factors.**

include policy-related challenges, financial barriers, issues at the facility level, and workforce challenges.

## Service Delivery and Organisation of Care

Service delivery challenges operated through a compound mechanism - inadequate clinical spaces combined with limited avenues for diabetes education created service gaps that systematically hindered comprehensive care delivery [35]. This mechanism was reinforced by the absence of standardised clinical guidelines, leading to inconsistent care provision and suboptimal outcomes [36]. Ugwu et al. found that most primary care physicians were unaware of any diabetes management guidelines, suggesting the severity of this issue [36]. Furthermore, the fragmentation of services and lack of teamwork undermined the quality and continuity of diabetes care, resulting in poor feedback, coordination, and management of patient transitions across levels of care [37].

## Health Workforce

Workforce challenges posed another significant barrier to effective diabetes care. Shortages of healthcare professionals, particularly in rural primary care facilities, hampered effective diabetes care delivery [38]. This shortage was exacerbated by the fact that frontline PHC workers often lacked up-to-date knowledge, skills, and training in evidence-based standards of

**Table 2. Summary of included studies and findings.**

| Author(s) | Country | Aim/Purpose of the Study | Study Participants | Key Findings |
|---|---|---|---|---|
| *Supply-Side Issues* | | | | |
| Ajisegiri WS, Abimbola S, Tesema AG, Odusanya OO, Peiris D, Joshi R. "We just have to help": Community health workers' informal task-shifting and task-sharing practices for hypertension and diabetes care in Nigeria. Front Public Health. 2023 Jan 26;11:1038062. | Nigeria | Explore roles and practices of CHWs in hypertension and diabetes care delivery. | CHWs (n=77), local and state government stakeholders (n=7) | **Macro**<br>Inadequate training for Community Health Workers (CHWs) (reported by 84% of CHWs)<br>Inadequate supplies of equipment (81%)<br>Poor infrastructure (71%)<br>**Meso**<br>Inadequate supervision of CHWs (52%)<br>CHWs flexibly implement national guidelines for hypertension and diabetes, exercising considerable discretion in interpretation.<br>Limited communication between Primary Health Care (PHC) facilities and higher-level facilities during referral processes<br>Inadequate medication supply at PHC facilities<br>Need for simplified NCD clinical algorithms/guidelines that CHWs can easily use at the point of care.<br>Continuous capacity building for CHWs is required to enhance their roles in NCD care.<br>**Micro**<br>Government stakeholders' rigid views on restricting CHWs' role to screening and referral due to insufficient formal training and limited medication supply<br>Balancing the need for referrals with maintaining patients' trust in the services provided at the PHC facility |
| Bosun-Arije FS, Ling J, Graham Y, Hayes C. Organisational factors influencing non-pharmacological management of type 2 diabetes mellitus (T2DM) in public hospitals across Lagos, Nigeria: A qualitative study of nurses' perspectives. Diabetes Research and Clinical Practice. 2020 Aug 1;166:108288. | Nigeria | Examine organisational factors influencing T2D management | Nurses from urban, sub-urban, and rural regions (n=17) | **Meso**<br>Issues of:<br>Information and knowledge management within the clinical environment<br>Relationship management within the clinical environment<br>Staffing levels and nurse-to-patient ratio<br>Decision-making autonomy of doctors<br>Availability of medical supplies such as glucometers and strips |

*(Continued)*

**Table 2.** (Continued)

| Author(s) | Country | Aim/Purpose of the Study | Study Participants | Key Findings |
|---|---|---|---|---|
| Iregbu SC, Duggleby W, Spiers J, Salami B. An Interpretive Description of Sociocultural Influences on Diabetes Self-Management Support in Nigeria. Global Qualitative Nursing Research [Internet]. 2022;9. Available from: https://www.scopus.com/inward/record.uri?eid=2-s2.0-85138718707&doi=10.1177%2f23333936221121337&partnerID=40&md5=49e6236d-3191543abb277dde2dbdec95 | Nigeria | To explore how the Nigerian social and cultural context influences healthcare providers' self-management support practices | HCPs from two diabetes clinics (n=19) | **Macro**<br>Lack of health insurance, leading to out-of-pocket payments for treatment, which is significant considering the average income of Nigerians<br>Inadequate funding for state-owned and -operated hospitals compared to federally funded hospitals, resulting in fewer staff and limited capacity to provide comprehensive care.<br>**Meso**<br>The rigidity of a hierarchical acute care model of diabetes services not designed for ongoing diabetes self-management education or support<br>Healthcare providers working in silos, with no evident interprofessional collaboration<br>The absence of designated diabetes educators in hospitals leading to unclear roles in providing diabetes education<br>Inadequate essential resources and leadership capacity in hospitals to effect change, resulting in healthcare providers feeling helpless and unable to make fundamental changes<br>Lack of support for continuing education for healthcare providers, resulting in a lack of skills for individualised support for patients<br>The absence of a structured education plan and uniform approach among healthcare providers<br>Absence of a structure for continued self-management support, patient follow-up, and coordination of care within the community<br>**Micro**<br>Patients' widespread beliefs in the supernatural origins of diabetes impacting self-management and support strategies and leading to delayed healthcare<br>Patients' inability to afford medical bills, drugs, and necessary resources for diabetes self-management forcing healthcare providers to adapt treatment<br>Patients' lack of understanding of diabetes and its management, even after attending the clinic for some time<br>Individual-family interdependence, myths, and limited understanding of diabetes and its management among patients |

*(Continued)*

**Table 2.** (Continued)

| Author(s) | Country | Aim/Purpose of the Study | Study Participants | Key Findings |
|---|---|---|---|---|
| Ugwu E, Young E, Nkpozi M. Diabetes care knowledge and practice among primary care physicians in Southeast Nigeria: a cross-sectional study. BMC Family Practice. 2020 Jul 1;21(1):128. | Nigeria | To evaluate diabetes care knowledge and practice among primary care physicians | Primary care physicians (n=64) | **Macro** <br> Absence of local diabetes clinical practice guidelines (CPG), identified as the most significant challenge faced by primary care physicians (PCPs) in managing people with diabetes. <br> **Meso** <br> Lack of access to diabetes specialists, identified by 70% of the respondents <br> Lack of access to allied healthcare professionals, identified by 66% of the respondents <br> Shortage of diabetes specialists and reliance on PCPs as the most considerable diabetes care medical workforce in Nigeria <br> Low level of diabetes care knowledge among PCPs in Southeast Nigeria, indicating a significant gap in the quality of primary health care for diabetes. <br> **Micro** <br> Low knowledge of glycaemic cut-offs for diagnosis of diabetes among PCPs: 26.6% for FBG, 45.3% for RBG, and 10.9% for A1c <br> Lack of knowledge about prediabetes among PCPs <br> Only 7.8% of PCPs correctly knew all three glycaemic values for diagnosis of diabetes, while 48.4% did not have correct knowledge of any one of the glycaemic cut-offs <br> 79.7% of respondents were not aware of any diabetes management guideline |
| Ugwu E, Onung S, Ezeani I, Olamoyegun M, Adeleye O, Uloko A. Barriers to diabetes care in a developing country: exploratory evidence from diabetes healthcare providers. Journal of Advances in Medicine and Medical Research. 2020;32(10):72–83. | Nigeria | To explore significant barriers and facilitators to diabetes care from the perspectives of diabetes healthcare providers | Diabetes healthcare providers (n=129) | **Macro** <br> Poverty (ranked first among the major barriers, reported by 89.1% of respondents) <br> Dysfunctional primary healthcare (PHC) system <br> Unregulated alternative medicine practices. <br> **Meso** <br> Lack of diabetes care support staff <br> **Micro** <br> Low diabetes awareness (reported by 82.9% of participants) <br> False religious and cultural beliefs <br> Poor general practitioner's (GP's) knowledge of diabetes care <br> Poor patient's knowledge of diabetes self-management |

(Continued)

**Table 2.** (Continued)

| Author(s) | Country | Aim/Purpose of the Study | Study Participants | Key Findings |
|---|---|---|---|---|
| *Mogre V, Johnson NA, Tzelepis F, Paul C. Attitudes towards, facilitators and barriers to the provision of diabetes self-care support: A qualitative study among healthcare providers in Ghana. Diabetes & Metabolic Syndrome: Clinical Research & Reviews. 2019 May 1;13(3):1745–51.* | Ghana | Explore attitudes, barriers, and facilitators to diabetes care support among HCPs | Healthcare providers (n=14) | **Macro**<br>Inadequate training of healthcare professionals (HCPs) in diabetes care due to little attention given to diabetes and its treatment in the curriculum<br>Inadequacy of continuous professional development programmes for HCPs to equip their competencies regarding diabetes care<br>Limited internet connectivity in hospitals hindering opportunities for HCPs to learn more about diabetes<br>Nutrition and dietetic care services are not insured, requiring patients to pay out-of-pocket<br>**Meso**<br>Poor inter-professional collaboration among the various types of healthcare professionals required to provide diabetes care (especially between nurses and nutritionists/dieticians)<br>Insufficient number of trained healthcare professionals in diabetes clinics<br>HCPs' perception of inadequate training in diabetes care<br>Lack of regular access to essential diagnostic tools to test for critical clinical variables such as HbA1c<br>**Micro**<br>HCPs adopting information-centric approaches instead of building patient-provider teams and patients' confidence in behaviour change<br>Good teamwork among HCPs of the same profession (e.g., nurses) but poor inter-professional collaboration<br>Some types of healthcare professionals feel more competent than others or perceive a more critical role in diabetes care, hindering inter-professional collaboration.<br>Language barriers between patients and HCPs affect effective communication, health literacy, and patients' trust in HCPs to manage their diabetes.<br>Patients' inability to pay for uninsured services like nutrition and dietetic care negatively affects the receipt of self-care support regarding diet. |

*(Continued)*

**Table 2.** (Continued)

| Author(s) | Country | Aim/Purpose of the Study | Study Participants | Key Findings |
|---|---|---|---|---|
| *Rn OEY, Rn OLY, Abimbola OO. Primary Health Care Nurses' Competencies and Resources Availability for Diabetes Mellitus Care at Local Government Areas of Ibadan. 2020, https://api.semanticscholar.org/ CorpusID:216561443* | Nigeria | To investigate PHC nurses' competency in diabetes mellitus care and evaluate the material resources available for care in the community | Primary healthcare nurses (n=88) | **Meso** Lack of availability of insulin and insulin syringes at 98.4% and 100% of the primary healthcare centres (PHCs), respectively. Unavailability of glucometers, glucometer test strips, and teaching aids at 53.2%, 58.1%, and 62.9% of the PHCs, respectively Lack of functioning refrigerators for storing insulin in 64.5% of the PHCs. **Micro** Nurses' knowledge gaps in diabetes mellitus care: a. 69.3% got the definition of diabetes mellitus diet wrong b. 71.6% did not know the best method for home glucose testing c. Only 27.3% knew the treatment for low blood glucose d. 62.5% did not know what could cause a low blood glucose level. 73.9% did not know the signs of ketoacidosis Overall, 58% of nurses had adequate knowledge of diabetes mellitus 36.4% of nurses had never identified or treated hyperglycaemia 54.5% of nurses practised below the expected level of diabetes mellitus nursing skills, while 45.5% practised within the expected level |
| *Demand-Side Issues* | | | | |
| *Okoronkwo IL, Ekpemiro JN, Okwor EU, Okpala PU, Adeyemo FO. Economic burden and catastrophic cost among people living with type2 diabetes mellitus attending a tertiary health institution in southeast zone, Nigeria. BMC Res Notes. 2015 Oct 1;8:527.* | Nigeria | To assess the magnitude of economic burden borne and catastrophic costs incurred by people living with type 2 diabetes mellitus | T2D patients (n = 308) receiving care outpatient care | **Macro** High cost of care impeding effective diabetes management due to low purchasing power, out-of-pocket payment at the point of need, and absence of prepayment mechanisms The magnitude of economic burden and catastrophic cost observed in the study area, despite being one of the oil-producing states with better resource allocation Implementation of the National Health Insurance Scheme, which presently covers some categories of government workers but does not adequately protect people experiencing poverty from the impact of out-of-pocket spending and increasing care costs (inequity) **Micro** Disproportionate experience of catastrophic costs tilting towards lower socioeconomic groups, indicating that people with low incomes are not protected from the impact of out-of-pocket spending and increasing care costs People living with diabetes (PLWD) need to contend with high costs of care throughout their lifetime |

*(Continued)*

**Table 2.** (Continued)

| Author(s) | Country | Aim/Purpose of the Study | Study Participants | Key Findings |
|---|---|---|---|---|
| Opare-Addo MNA, Osei FA, Buabeng KO, Marfo AF, Nyanor I, Amuzu EX, et al. Health-care services utilisation among patients with hypertension and diabetes in rural Ghana. African Journal of Primary Health Care & Family Medicine [Internet]. 2020 [cited 7AD Jan 1];12(1). Available from: http://www.phcfm.org/index.php/PHCFM/article/view/2114 | Ghana | To determine the prevalence of hypertension and diabetes in rural districts and factors influencing preferences of healthcare facilities/ services | Patients from rural districts (n=684) | **Macro** Educational level and health insurance enrolment increased the likelihood of seeking care in public hospitals. **Meso** The perceived sound quality of care influenced the preference for health facilities. A little above half of the respondents (73; 53.28%) cited quality of care as the main reason for their preferred health facility, followed by proximity/distance, which 48 (35.04%) respondents also cited. **Micro** Predisposing factors (age, marital status, education, occupation) and enabling factors (income, socioeconomic status) influenced preference for health facility |
| Kretchy IA, Koduah A, Ohene-Agyei T, Boima V, Appiah B. The Association between Diabetes-Related Distress and Medication Adherence in Adult Patients with Type 2 Diabetes Mellitus: A Cross-Sectional Study. J Diabetes Res. 2020 Mar 1;2020:4760624. | Ghana | To estimate diabetes-specific distress and assess its impact on medication adherence | T2D patients recruited from outpatient clinic (n=188) | **Macro** Lack of psychological and social support systems for diabetes-related distress **Meso** Lack of routine screening for diabetes-related distress in the health system **Micro** High levels of diabetes-related distress associated with lower medication adherence, negative emotions, dietary concerns, dissatisfaction with external support, and diabetes management helplessness |
| BeLue R, Diaw M, Ndao F, Okoror T, Degboe A, Abiero B. A cultural lens to understanding daily experiences with type 2 diabetes self-management among clinic patients in M'bour, Senegal. Int Q Community Health Educ. 2012 2013;33(4):329–47. | Senegal | To examine experiences with diabetes self-management among clinic patients using the PEN-3 cultural model | Clinic patients with type 2 diabetes (n=54) | **Macro** Lack of integration between Western and Traditional medicine, with participants having their own perspectives about the utility of Traditional and/or biomedical/Western medicine Lack of health insurance systems to offset costs of medications High unemployment rate/challenges obtaining necessities like food Limited access to healthy foods due to cost **Meso** Family dynamics serving as both supportive and inhibiting forces **Micro** Belief in the supernatural origin of diabetes, individual-family interdependence, myths and limited understanding of diabetes, financial challenges related to accessing medical care and adhering to prescribed diabetic diet. |

*(Continued)*

**Table 2.** (Continued)

| Author(s) | Country | Aim/Purpose of the Study | Study Participants | Key Findings |
|---|---|---|---|---|
| *Mogre V, Johnson NA, Tzelepis F, Paul C. Barriers to diabetic self-care: A qualitative study of patients' and healthcare providers' perspectives. Journal of Clinical Nursing. 2019;28(11–12):2296–308.* | Ghana | Explore patient and provider perspectives on diabetes care barriers | Healthcare providers (n=14) and people living with type 2 diabetes (n=23) | **Macro**<br>Low income levels, inadequate access to a variety of foods<br>Poor health insurance coverage for diabetes care<br>Limited availability of healthcare providers trained in diabetes care<br>**Meso**<br>Subjective norms/normative beliefs:<br>a. Inadequate family support<br>b. Culture and beliefs<br>c. Social stigma<br>**Micro**<br>Attitudes/behavioural beliefs: a. Misconception and use of herbal medicine<br>b. Difficulty changing old habits<br>c. Intentional nonadherence and fatalism<br>d. Worrying about the continuous taking of medication.<br>e. Feeling lazy to perform exercise<br>f. Side effects of medication<br>Perceived behavioural control/control beliefs:<br>a. Nonreceipt of self-care support to perform self-monitoring of blood glucose (SMBG)<br>b. Inadequate knowledge and skills to operate the glucometer<br>c. Patients having comorbid conditions<br>d. Low income levels<br>e. Inadequate access to a variety of foods<br>f. Diet recommendations being too restrictive<br>g. Busy work schedules |

**Table 3 .  Summary of Key Findings on Diabetes Care in West Africa by Health System and Contextual Domains.**

|  | Key Findings | Source Studies |
|---|---|---|
| Service Delivery | • Inadequate clinical spaces for diabetes care<br>• Limited avenues for diabetes education and counselling<br>• Absence of standardised clinical guidelines | [35–37] |
| Health Workforce | • Shortages of healthcare professionals, especially in rural areas<br>• PHC workers lack current knowledge and skills | [38–40] |
| Health Financing | • Inadequate health insurance coverage<br>• High out-of-pocket expenditures<br>• Inadequate funding for state-owned hospitals | [38,39,41,42] |
| Governance | • Poor prioritisation of chronic diseases in health agendas<br>• Inadequate robust regulatory framework for diabetes care | [39,41] |
| Social Determinants of Health | • Widespread beliefs in supernatural causes of diabetes<br>• Cultural food preferences challenging dietary management<br>•Education, proximity of health facilities influencing utilisation | (36,39,42,44–46) |
| Self-Management Support | • Family and social support as critical enablers<br>• Women's central role in health promotion<br>• Psychological strains of diabetes and its negative effects on management | [41,44,46] |

diabetes care, leading to poor management of patients [39,40]. Oyewole et al. observed negative attitudes towards caring for people with diabetes among care providers, further impacting the quality of care provided [40]. However, the reviewed studies did not explicitly explore the underlying reasons for these negative attitudes.

The need for coordination and collaborative practices among healthcare provider cadres compounded these workforce issues. Instead of adopting an integrated, consultative approach to patient care, providers often worked in silos [43]. The lack of clear role assignments and poor interprofessional collaboration among different types of healthcare providers involved in diabetes care, such as doctors, nurses, dietitians, and health educators, resulted in these noticeable silos. Iregbu et al. [43] found that nurses felt they needed to be more utilised and that their roles needed to leverage their knowledge and skills optimally. Some doctors believed they were solely responsible for providing diabetes education with little contribution from other professionals. Referrals from doctors to dietitians needed to be more consistent, and doctors sometimes needed to remember to refer patients or believed they could provide dietary counselling themselves.

Similarly, Mogre et al. showed that silos existed between nurses, doctors, and dietitians/ nutritionists. There needed to be more teamwork and cooperation across these professional groups, with more collaboration occurring within specific professionals (e.g., among nurses) rather than between different health personnel. Poor communication, lapses in sharing information, competition, and perceptions of relative importance in diabetes care contributed to these silos. These professional silos primarily occurred at the provider level, reflecting a lack of integrated, multidisciplinary care teams.

## Health Financing

Financial barriers operated through a clear mechanism - the lack of health insurance and high out-of-pocket expenditures created a direct pathway that limited access to quality diabetes care [38,39,41,42]. Okoronkwo et al. noted the inadequate implementation of the National Health Insurance Scheme in Nigeria, which failed to protect people experiencing poverty from the impact of out-of-pocket spending and increasing care costs [41]. This issue was worsened by inadequate funding for state-owned and operated hospitals compared to federally funded

hospitals, leading to fewer staff and limited capacity to provide comprehensive diabetes care [43]. In some cases, crucial components of diabetes self-management support, such as nutrition and dietetic care, were not covered by health insurance schemes, forcing patients to pay out-of-pocket and further limiting access to comprehensive care [35].

## Leadership/Governance

Several policy and governance issues undermined the provision of quality diabetes care. Authorities poorly prioritised and planned for the growing burden of chronic diseases like diabetes as health agendas continued to be dominated by infectious diseases [38]. This lack of prioritisation was further worsened by high poverty rates and inadequate financial risk protection policies that limited the affordability of care [41,42]. Consequently, the health system lacked the necessary resources and preparedness to manage the increasing prevalence of diabetes effectively.

Another significant challenge was the lack of a robust regulatory framework to ensure evidence-based practices and maintain quality standards in diabetes care. Ugwu et al. reported unregulated alternative medicine practices in Nigeria, which hindered effective diabetes management by promoting the use of unproven or potentially harmful treatments.

The review identified significant evidence of challenges in service delivery, health workforce, financing, and governance in primary healthcare settings in West Africa. However, there were noticeable gaps in evidence regarding health information systems and comprehensive access to essential medicines and technologies for diabetes care. However, some studies found shortages in necessary resources and supplies, such as glucometers, test strips, and teaching aids, which hindered effective diabetes management [40,43].

## Contextual Factors

Cultural beliefs functioned through a distinct behavioral mechanism where widespread beliefs in supernatural causes of diabetes, such as curses and spiritual attacks [38,39,43,44], created a pathway that led patients first to traditional healers before seeking formal healthcare. This mechanism was reinforced when patients perceived positive outcomes from traditional medicine, further delaying engagement with evidence-based therapies [38].

Furthermore, the dominance of carbohydrate-rich staple foods and dishes in typical West African cuisine presented a significant challenge for the dietary management of diabetes [39,44]. The high consumption of carbohydrate-rich foods in West African diets was due to cultural preferences, the affordability of these staple foods compared to healthier options, and the wide availability of high-carbohydrate ingredients in local markets and cuisine. These factors posed significant challenges for adopting diabetes-friendly diets in these contexts.

Distance to health facilities was also cited as a factor influencing the choice of healthcare providers, with a significant proportion of respondents more likely to utilise facilities nearby [45].

Interestingly, while some cultural beliefs posed barriers, family and social support systems emerged as critical enablers deeply rooted in sociocultural norms. Women were identified as playing a central role in leading household dietary modifications for diabetes management, leveraging their traditional responsibilities in food preparation and family caregiving [43]. However, the impact of family dynamics was not uniformly positive. The lack of social support also contributed to treatment nonadherence and difficulties implementing lifestyle changes in specific contexts (44). These findings emphasised the interplay of cultural beliefs, gender roles, family structures and their influence on diabetes care.

At the behavioural level, patients' preferences for traditional medicine over biomedical care and delays in care-seeking (albeit often related to high costs of care) were found to be significant barriers to effective diabetes management [43]. Crucially, mental health emerged as a critical psychosocial determinant shaping health outcomes for diabetes. The study by Kretchy et al. highlighted the multifaceted nature of diabetes-related distress and its strong association with poorer self-management behaviours and decreased medication adherence in the Ghanaian context.

The psychological impact operated through a bidirectional mechanism in which diabetes-related distress led to poor treatment adherence, which in turn worsened health outcomes, creating a self-reinforcing cycle [46]. This mechanism was amplified by inadequate social support systems and limited access to mental health services.

Notably, the study shed light on how people living with diabetes experience psychological distress due to poor social support, concerns related to meeting dietary requirements, and the risks of developing complications, leading to significant psychological turmoil. The psychological impact thus extended beyond individual concerns to include socioeconomic well-being. Furthermore, dissatisfaction with external support from healthcare providers and social networks was found to be a significant component of diabetes distress. Patients who did not receive support from their caregivers and care providers were more likely to experience distress and have poorer medication adherence.

These findings demonstrate how multiple mechanisms - financial, structural, cultural, and psychosocial - interact to influence diabetes care in West Africa. The interaction between these mechanisms often creates compound effects: for instance, when financial barriers intersect with cultural beliefs to reinforce delayed care-seeking, or when professional silos combine with psychosocial factors to impact care quality. Understanding these interactive mechanisms provides crucial insights for developing targeted interventions.

## Discussion

### Transforming Diabetes Care - Insights from a PHC-Centered Analysis

Amidst the rising burden of NCDs in LMICs, our scoping review explored the factors influencing access to, utilisation of, and quality of diabetes care in West African primary health systems. This review catalogues barriers and facilitators and unravels the intricate connections between health systems, sociocultural contexts, and individual experiences that shape diabetes care, highlighting the importance of deeply rooted cultural beliefs and the lived experiences of both patients and providers.

As we discuss results, we aim to map out a future that recognises the uniqueness of the West African context while drawing on other, mostly similar global contexts to improve diabetes care in resource-constrained settings.

Our analysis of diabetes care in West Africa is grounded in the synthesis of three complementary theoretical frameworks: the WHO Operational Framework for Primary Health Care (PHC), the Social Determinants of Health (SDH) model, and the Innovative Care for Chronic Conditions (ICCC) model. This multi-framework approach provides a comprehensive lens through which we interpret our findings and develop recommendations. By leveraging the strengths of each framework, we create a more robust analytical tool for understanding and addressing diabetes care in West Africa.

This synthesis allows us to capture the complexity of diabetes care in LMICs, addressing challenges such as fragmented health systems, the impact of social determinants on diabetes management, and the integration of traditional healing practices. Our findings suggest that when adapted to LMIC contexts, these frameworks can provide a more in-depth

understanding of diabetes care challenges. For instance, the ICCC's emphasis on patient and family involvement is enriched by considering the gendered aspects of care prevalent in West African contexts.

The multi-framework approach enables us to interpret our results using established theoretical constructs and contributes to the evolution of these frameworks for better applicability in diverse global health contexts.

## Strengthening Integrated Health Services for Diabetes Care – The Challenges, The Opportunities

Our review reveals profound health system barriers impacting diabetes care in West Africa. Inadequate infrastructure, lack of standardised guidelines, and fragmented services compromise access to and quality of diabetes care. Workforce shortages, insufficient training, and financial barriers further exacerbate these challenges. Overarching these issues are policy gaps, with chronic diseases often deprioritised. Despite these challenges, we uncover critical leverage points to improve diabetes care, potentially leading to a fundamental change in resource-constrained settings.

## Addressing Workforce Challenges and Care Fragmentation

In West Africa, the 'ideal' integrated PHC system is unique. Healthcare providers often navigate the intersection of biomedical approaches and traditional healing practices in diabetes management. This dual paradigm necessitates reimagining patient education and adherence support rooted in local sociocultural structures.

Although West Africa's health system barriers share commonalities with other LMICs, they manifest in ways that challenge conventional global health models. For instance, workforce shortages transcend numerical deficits, encompassing capacity issues and competing health priorities. The persistent tension between addressing infectious diseases and the rising burden of NCDs exerts unique pressure on the health system, necessitating strategic resource reallocation and re-evaluations of care delivery. This dynamic connection between communicable and noncommunicable disease management represents a blind spot in many global frameworks for chronic care [25]. Similar challenges in resource reallocation from infectious disease programs to NCD management are evident in Southeast Asian countries like Indonesia and Vietnam [47], highlighting a broader LMIC phenomenon.

Based on our findings, care fragmentation in West Africa reveals a misalignment between long-term diabetes care needs and current primary health service structures. This misalignment, compounded by resource constraints limiting access to essential medicines and diagnostics, creates a system ill-equipped for effective diabetes management. Moreover, fragmented referral systems further worsen this issue, hindering the provision of comprehensive and continuous care for diabetes patients. Similar challenges have been observed in other regions, like India and South Africa [48,49], where efforts to integrate diabetes care into primary health services have faced implementation barriers, particularly in coordinating care across different health system levels and contributing to poor health outcomes.

The ICCC model's emphasis on prepared, proactive practice/care teams requires substantial reinterpretation in regions like West Africa. Preparedness in these regions extends beyond clinical knowledge to encompass cultural competence and resource-constrained problem-solving. This calls for rethinking health workforce training and fostering adaptability and innovation beyond standardised guidelines. Addressing diabetes care in LMICs demands more than merely adapting existing care models. It requires a complete rethinking of delivering chronic care, considering local realities, and strategically using global best practices.

## Adapting Global Best Practices to Local Contexts

While global recommendations and best practices provide valuable guidance for diabetes care, our findings assert the critical importance of translating these into locally relevant and feasible interventions. This adaptation process must consider West African countries' unique challenges, resources, and sociocultural contexts. For instance, while comprehensive, the WHO's Package of Essential Noncommunicable (PEN) disease interventions may need significant modification to fit the West African context. Our review highlighted the prevalence of beliefs in supernatural causes of diabetes, the utilisation of traditional medicine and the dominance of carbohydrate-rich diets, necessitating culturally sensitive adaptations.

Adapting the PEN package could involve several vital strategies. Diabetes education programs could acknowledge traditional beliefs while integrating biomedical explanations. Additionally, nutritional guidance could focus on modifying traditional West African dishes to be more diabetes-friendly. This approach is similar to efforts in the USA to adopt the DASH diet for African American communities [50]. Furthermore, the decisive role of family support systems identified in our review suggests that family-centred diabetes education could be more effective than individual-focused approaches, aligning with successful community-based interventions in other developing regions [51].

The adaptation process must be context-specific and involve critical elements such as stakeholder engagement, pilot testing, and continuous evaluation. Including patients, healthcare providers, traditional healers, and community leaders in the adaptation process ensures relevance and acceptability. Implementing adapted interventions on a small scale allows for assessing feasibility before a broader rollout. Regular evaluation of the effectiveness of these adapted interventions will enable iterative improvements over time.

Deficiencies in the health system stem from direct issues, such as workforce shortages and inadequate supplies, which hinder service delivery. Additionally, indirect factors, such as fragmented care coordination, result in discontinuity of care and poor health outcomes. These system-level challenges become particularly potent when they intersect with sociocultural factors.

## Multisectoral Policy and Action, People and Communities – The Broader Determinants

While addressing health system barriers is crucial, our review uncovered the profound impact of factors beyond the health sector on diabetes care in West Africa. A multisectoral lens reveals how finance, education, urban planning, and agriculture policies significantly influence diabetes care.

For instance, our review revealed that the lack of financial risk protection mechanisms, such as comprehensive health insurance schemes, directly correlates with reduced access to and utilisation of diabetes care and, hence, poorer health outcomes. This misalignment between health system structures and socioeconomic realities undermines broader health equity and universal health coverage aspirations [52,53]. LMICs face unique hurdles due to high poverty rates and limited formal employment sectors, which complicate the implementation of health insurance models [54].

Our review highlights several promising examples of intersectoral collaboration, demonstrating the transformative potential of coordinated action. The implementation of "sin taxes" on unhealthy products in the Philippines, ingeniously generating revenue for health programs, including diabetes prevention, stands out as a beacon of innovative policy [55]. Equally compelling is Rwanda's expansion of community-based insurance schemes, which have significantly reduced out-of-pocket expenses and improved healthcare access for vulnerable populations [56]. These initiatives emphasise that successful Universal Health Coverage requires collaboration across sectors to establish sustainable financing

Education, housing, and transportation sectors play crucial roles in creating environments that promote or hinder healthy lifestyles. The "Ciclovia" program in Colombia exemplifies how urban planning can encourage physical activity and offers an adaptable model for LMICs [57,58]. Such initiatives transform communities into active contributors to health promotion.

As we shift our focus to empowered people and communities, we must rethink the connection between health systems and those they serve. The success of interventions at the system level is tied to their capacity to involve and empower individuals and communities as active partners in preventing and managing diabetes. The connection between system factors and community empowerment requires a comprehensive, system-based approach to enhance diabetes care in LMICs. This approach should acknowledge the interdependence of all PHC components and utilise local strengths and cultural resources to develop sustainable, contextually suitable solutions.

## Empowering People and Communities

Our exploration of cultural beliefs and traditional medicine practices provides further insight into our second research question, illustrating how these factors shape diabetes care. The widespread influence of cultural norms and alternative medicine on care-seeking behaviours demonstrates the intricate relationship between sociocultural contexts and health system utilisation.

## Cultural Beliefs and Traditional Medicine

The influence of cultural beliefs, social norms, and complementary and alternative medicine on diabetes care-seeking behaviours in Africa is widely recognised [59,60]. Beliefs in supernatural causes of illness often lead patients to seek traditional healers before seeking formal healthcare. When these beliefs interact with health system limitations, they create reinforcing cycles that delay care-seeking and further erode trust in formal healthcare systems. Rather than viewing these factors as obstacles, they present unique opportunities for creating culturally resonant care models. The systematic study by Zeh et al. [61] highlights the significance of cultural relevance in diabetes care and pushes us to rethink this in various contexts.

Cultural sensitivity remains crucial even in situations of trans-migration. A study of Korean Americans showed improved glycaemic control when community health workers had culturally tailored training [62]. This example calls for integrating cultural competence into healthcare delivery, linking biomedical and traditional care systems.

Studies in Tanzania and ASEAN countries further highlight the role of traditional medicine in diabetes management, emphasising the need for culturally sensitive interventions [63,64]. These indicate the importance of developing culturally appropriate health promotion strategies that engage community stakeholders to raise awareness about diabetes and promote timely care-seeking.

By leveraging cultural beliefs and traditional practices, we can empower communities to take ownership of diabetes prevention and management, fostering sustainable, context-appropriate solutions.

## Mental Health and Psychosocial Aspects of Diabetes Care

While cultural factors significantly influence diabetes care, our review also reveals the critical dimension of mental health, particularly in LMIC settings such as West Africa. The manifestation of psychological vulnerability as burnout, low motivation, and disengagement from treatment [65] points to a cyclical pattern where poor mental health further deteriorates glycaemic control. The psychological impact of diabetes extends far beyond the burden of

self-care routines, encompassing complex psychosocial dimensions that are often overlooked in traditional care models.

Our review found that diabetes distress is deeply intertwined with psychosocial factors and healthcare realities. Kretchy et al. revealed how diabetes can negatively impact patients' emotional well-being, social interactions, and ability to manage their condition effectively. The study identified four key areas of distress: negative emotions about diabetes, dietary concerns and diabetes care, dissatisfaction with external support, and diabetes management helplessness. These factors were found to influence medication adherence significantly, highlighting the complex interplay between psychological, social, and clinical aspects of living with diabetes [46]. Psychological distress not only directly impacts treatment adherence but often initiates a cascade where poor adherence leads to worse outcomes, which in turn increases the psychological burden. This cycle is particularly challenging in contexts where mental health support is limited or stigmatised.

Similar findings in other cultural contexts, such as among Hispanic men in the US [66], highlight the universal nature of this challenge. Moreover, the psychological burden often extends to the entire family unit. Diabetes-related financial stress significantly impacts family dynamics and the mental health of both patients and caregivers [67].

A growing body of evidence asserts a bidirectional relationship between diabetes and mental health disorders [68,69], with comorbidities significantly impairing self-management behaviours [70]. The chronic nature of diabetes and its associated complications contribute to substantial psychological distress and diminished quality of life [71,72], hence creating a cycle of deteriorating physical and mental health.

Addressing the mental health aspects of diabetes care is crucial for empowering individuals to manage their condition effectively, emphasising the need for holistic, person-centred approaches to diabetes care.

## Gender Dynamics in Diabetes Management

Just as mental health plays a crucial role in diabetes management, our review also uncovered significant gender dynamics influencing care. Women's central role in leading household dietary modifications for diabetes management, leveraging their traditional responsibilities in food preparation and family caregiving, is a crucial finding resonating across many cultures, particularly in Africa and other LMICs [43]. This observation serves as a springboard for understanding the interactions between gender and diabetes care.

Women's pivotal role in dietary management significantly influences household nutrition, shaped by local cultural norms and socioeconomic factors [73]. However, this responsibility is a double-edged sword: It empowers women to lead health-promoting changes and adds to their substantial caregiving burden. Research in South Africa and among Black and Latina women in the US highlights how women with diabetes often struggle to balance their own health needs with caregiving roles [74,75].

Economic constraints significantly impact diabetes management for women in Africa. Studies reveal that low income levels undermine adherence to medication, dietary recommendations, and self-monitoring practices [74,76]. Women often prioritise public healthcare despite its limitations, as they struggle to afford private care, essential monitoring tools, and diabetes-friendly foods. These financial barriers frequently force women to compromise on their diabetes care, potentially leading to poorer health outcomes. This struggle is exacerbated by competing household demands and social expectations around food preparation and consumption – a pattern also observed in other LMICs like India [77].

While gender roles, particularly women's central influence in household dietary decisions and caregiving, can potentially enhance diabetes management, this benefit is undermined by systemic barriers such as financial constraints and limited access to healthy foods.

The impact of cultural beliefs and gender roles on diabetes management illustrates the emphasis of the SDH model on sociocultural factors affecting health. These profoundly ingrained factors in West African societies highlight the importance of diabetes care strategies that consider both the clinical and broader social context of health.

In essence, the provision of diabetes care in LMICs is influenced by factors that involve community-level dynamics and broader systemic issues - an intricate web of interactions at macro, meso, and micro levels. For example, macro-level health financing policies directly impact meso-level health system capacities, affecting micro-level patient experiences. Resource scarcity at the meso level influences individual health-seeking behaviours, often causing patients to postpone seeking care or explore alternative treatments.

This analysis highlights the drawbacks of isolated approaches or solutions. It stresses the importance of comprehensive strategies that tackle policy obstacles, enhance healthcare system capabilities, and consider sociocultural factors to bring about positive improvements in diabetes care at all levels in West Africa.

## Research Gaps

Our scoping review reveals critical knowledge gaps in understanding and improving diabetes care in West Africa. The current evidence base, largely drawn from Nigeria, Ghana, and Senegal, leaves much of the region's diverse health systems and cultural contexts unexplored. This geographic limitation significantly constrains our understanding of how different health system structures and sociocultural factors influence diabetes care across the region.

Implementation research presents a particular blind spot. While our review identified numerous barriers and facilitators, evidence regarding effective interventions remains sparse. There appears to be a lack of indepth insight into strategies for integrating traditional healing practices, strengthening referral systems, and implementing task-shifting approaches in resource-constrained settings. Similarly, though financial barriers emerged as a crucial challenge, evidence on cost-effective care models and sustainable financing mechanisms appears limited.

The literature also reveals a concerning imbalance, heavily favouring provider and system-level perspectives over patient experiences. Understanding the lived experiences of people with diabetes, their care preferences, and the influence of family dynamics on diabetes management is crucial for developing culturally resonant interventions. This gap extends to quality measurement, where contextually appropriate metrics and benchmarks for West African settings are notably absent.

Two emerging areas require particular attention. First, despite the region's growing digital connectivity, evidence on effective digital health interventions for diabetes care remains limited. Second, while the importance of intersectoral action is widely acknowledged, research on implementing and evaluating multisectoral approaches is scarce. Understanding how different sectors can effectively collaborate to support diabetes care could unlock new possibilities for comprehensive care delivery.

Addressing these knowledge gaps through methodologically robust studies will be essential for developing sustainable approaches to diabetes care that reflect West African realities and resources. Future research must move beyond identifying problems to testing and evaluating solutions that can work within local contexts.

## The Path Forward

The path forward for diabetes care in West Africa and other LMICs demands a bold reimagining of health systems and societal structures, transcending traditional boundaries. This calls for a revolutionary approach to interweaving health system strengthening, community

empowerment, and multisectoral collaboration. At its core, this approach aligns with the WHO building blocks and the PHC framework while adapting to the unique challenges and opportunities in resource-constrained settings.

Central to this vision is strengthening health service delivery through innovative, context-specific strategies. We must amplify task-shifting approaches and implement a tiered strategy that trains community health workers in diabetes screening, education, and management while enhancing primary care providers' knowledge and skills through specialised curricula and programs in areas such as dietary management of diabetes. This helps address immediate workforce shortfalls and builds a system of diabetes care expertise. Additionally, we must develop and implement context-specific clinical guidelines, drawing inspiration from successful programs like the PACK (Practical Approach to Care Kit) adopted in South Africa, Botswana, and Brazil [78–80].

These strategies should be complemented by integrating diabetes care with other primary health services, including mental health support, and aligning with global trends towards holistic care models, as seen in places like Chile [81]. It is crucial to Train healthcare providers to recognise and address the psychosocial aspects of diabetes. We propose developing integrated care models incorporating mental health and traditional healing practices, ensuring evidence-based and culturally resonant care. A robust referral system leveraging digital health technologies should be a key component, ensuring continuity of care across different levels of the health systems.

The CASALUD model from Mexico offers valuable lessons in leveraging technology and community resources to improve diabetes care in resource-limited settings [82]. Building on this, we recommend developing integrated diabetes information systems that combine clinical, community, and social data. This system should support patient management, population health strategies, resource allocation decisions, and policy evaluation, facilitating multisectoral action through interoperability with other sectoral information systems. Furthermore, access to essential medicines and technologies for diabetes care must be improved through strengthened supply chain management. By harnessing digital technologies for improved inventory tracking and forecasting, we can ensure a consistent supply of essential diabetes medicines and diagnostics. This technological integration aligns with the broader vision of leveraging innovation to overcome resource constraints.

We must rethink capacity building as a collaborative endeavour spanning multiple sectors, including education, agriculture, urban planning, information technology, finance, and social services [83]. Such a multisectoral symphony can potentially redefine diabetes care. It can shift our focus from treating a disease to nurturing health-conscious societies. Envision communities where urban planners design cities that nudge citizens towards healthier lifestyles, where agricultural policies cultivate not just crops but community health, and where educational curricula sow the seeds of health literacy from early childhood.

The path forward also demands a fundamental shift in the power dynamics of healthcare, repositioning patients and communities from passive recipients to active co-creators of health. This transformation involves integrating culturally sensitive mental health support into diabetes care, implementing family-centred approaches that address the broader impact of diabetes on household dynamics and economics, and developing community-based support programs to mitigate social and economic challenges, especially for vulnerable populations like women. Engaging stakeholders through community-based education programs and peer support groups is essential. Moreover, exploring collaborative partnerships with traditional healers can foster culturally sensitive care delivery, bridging the gap between formal healthcare systems and community practices.

Strengthening the health information system is crucial to supporting these initiatives. Establishing clear indicators and targets and promoting data sharing across sectors will enable

evidence-based decision-making and policy influence. Innovative tools like the web-based ALMA Scorecards, successfully used for other health issues in developing countries [84], can be adapted to track progress in diabetes care. Robust monitoring and evaluation mechanisms will be crucial to the success of multisectoral initiatives.

Leadership and governance play a critical role in realising this vision. Effective multi-sectoral action for diabetes care in West Africa requires strong political commitment and stakeholder engagement. By prioritising diabetes and, by extension, NCDs on national health agendas, fostering cross-sectoral partnerships, and exploring innovative funding sources, LMICs can lay the foundation for more resilient and responsive PHC systems.

## Strengths and Limitations

The predominance of studies from Nigeria in our review may limit the generalizability of findings across West Africa due to contextual variations. Secondly, the heterogeneity of study designs, populations, and settings could introduce variability and impact comparability. Additionally, the lack of formal quality assessment of included studies, consistent with the scoping review methodology, may affect the reliability of the synthesised evidence. The focus on published literature risks potential publication bias, while the time-bound evidence synthesis may have missed developments in the rapidly evolving diabetes care landscape. Despite these limitations, this review aimed to offer a comprehensive synthesis, identifying critical gaps and highlighting areas for future research to advance strategies for improving diabetes primary care in West African settings.

## Conclusions

Our multi-framework analysis revealed the complex landscape of diabetes care in West Africa, calling for a significant change in how we conceptualise and implement health interventions in resource-constrained settings. Our findings highlight the need to move beyond siloed approaches and embrace a systems-thinking approach that recognises the connections between health systems, sociocultural contexts, and individual experiences. This holistic perspective requires us to redefine NCDs not just as medical conditions but also as sociocultural phenomena deeply ingrained in societies.

Looking ahead, the challenge lies in translating these insights into actionable strategies. This necessitates boldly reimagining health system strengthening efforts that prioritise context-specific innovations, empower communities as active partners in health creation, and foster multisectoral collaboration. Future research and policy initiatives must focus on developing and evaluating adaptive, culturally resonant models of care that can effectively navigate biomedical and sociocultural factors shaping diabetes management in LMICs. By embracing this complexity and leveraging local strengths, we can pave the way for more resilient, equitable, and effective NCD care systems that improve health outcomes and contribute to broader socioeconomic development.

## Supporting information

**S1 Protocol. Scoping Review Protocol.**
(PDF)

**S1 Checklist. PRISMA ScR Checklist.**
(DOCX)

**S1 Appendix. Sample Search Strategy - PubMed.**
(DOCX)

## Author contributions

**Conceptualization:** Abdul-Basit Abdul-Samed, Yasmin Jahan, Dina Balabanova, Tolib Mirzoev, Irene Akua Agyepong.

**Formal analysis:** Abdul-Basit Abdul-Samed, Yasmin Jahan, Veronika Reichenberger, Mary Pomaa Agyekum, Irene Akua Agyepong.

**Funding acquisition:** Tolib Mirzoev, Irene Akua Agyepong.

**Methodology:** Yasmin Jahan, Veronika Reichenberger, Mary Pomaa Agyekum, Henry Lawson, Irene Akua Agyepong.

**Supervision:** Dina Balabanova, Tolib Mirzoev.

**Validation:** Mary Pomaa Agyekum.

**Writing – original draft:** Abdul-Basit Abdul-Samed.

**Writing – review & editing:** Yasmin Jahan, Veronika Reichenberger, Ellen Barnie Peprah, Henry Lawson, Dina Balabanova, Tolib Mirzoev, Irene Akua Agyepong.

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
