## [Decision Letter · Decision Letter 0]

19 Sep 2024

PGPH-D-24-02021

Improving Type II Diabetes Care in West Africa: A Scoping Review of Barriers, Facilitators and the Way Forward

Dear Dr. Abdul-Samed,

Thank you for submitting your manuscript to PLOS Global Public Health. After careful consideration, we feel that it has merit but does not fully meet PLOS Global Public Health’s publication criteria as it currently stands. Therefore, we invite you to submit a revised version of the manuscript that addresses the points raised during the review process.

Considering the prevalence of Type-II Diabetes; this manuscript has merits and is very relevant for policy makers and also setting up the background for further research for better patient care. 

But, this manuscript in present form needs minor revision(minor clarifications and grammatical); as aptly highlighted by both the reviewers. 

We look forward to receiving your revised manuscript.

Kind regards,

Abhiruchi Galhotra, MBBS, MD

Academic Editor

Journal Requirements:

Additional Editor Comments (if provided):

Reviewers' comments:

Reviewer's Responses to Questions

**Comments to the Author**

1. Does this manuscript meet PLOS Global Public Health’s publication criteria ? Is the manuscript technically sound, and do the data support the conclusions? The manuscript must describe methodologically and ethically rigorous research with conclusions that are appropriately drawn based on the data presented.

Reviewer #1: Yes

Reviewer #2: Yes

2. Has the statistical analysis been performed appropriately and rigorously?

Reviewer #1: N/A

Reviewer #2: N/A

3. Have the authors made all data underlying the findings in their manuscript fully available (please refer to the Data Availability Statement at the start of the manuscript PDF file)?

Reviewer #1: Yes

Reviewer #2: Yes

4. Is the manuscript presented in an intelligible fashion and written in standard English?

Reviewer #1: Yes

Reviewer #2: Yes

5. Review Comments to the Author

Reviewer #1: In the PRISMA chart, provide clear reasons for excluding each article.

In the Result section: Add references for each study included in the table and also include the study design of the selected articles.

The articles should be organised into supply-side and demand-side issues, with a balanced mix of both.

The same flow should be followed in the discussion section.

Reviewer #2: The manuscript on Scoping Review of DM is written well yet it requires some minor changes, which are highlighted in the pdf file attached with it. Kindly check it from the pdf and address the minor issues.

6. PLOS authors have the option to publish the peer review history of their article (what does this mean? ). If published, this will include your full peer review and any attached files.

**Do you want your identity to be public for this peer review?** For information about this choice, including consent withdrawal, please see our Privacy Policy .

Reviewer #1: No

Reviewer #2: **Yes: ** Sanjana Agrawal

---

## [Author Response · Author response to Decision Letter 1]

3 Nov 2024

Abhiruchi Galhotra, MBBS, MD

Academic Editor

Dear Dr Abhiruchi,

Thank you for considering our manuscript for publication in PLOS Global Public Health, and we appreciate the time and effort of the reviewers whose feedback has been valuable. We have carefully revised the manuscript accordingly, and in the table below include point-by-point responses and actions taken to address each comment.

Reviewer 1 Response Page, (Lines)

"This should be moved to result section and the reason for exclusion of articles should be clear"

As suggested, we have moved the statement "Twelve papers were included in this scoping review after screening 4,290 documents" to the results section. We have also added a brief explanation of the reasons for article exclusion to clarify our selection process:

“Articles were excluded if they were not conducted in West African countries, did not focus on primary care for diabetes, or failed to report on factors affecting access, utilisation, or quality of care. We also excluded non-empirical articles. This rigorous exclusion process ensured that only relevant empirical studies addressing our specific research questions were included in the final analysis.” 9, (168 – 172)

"mention the (n=?) each study design included" We have addressed this in our response to a similar comment from Reviewer #2. The revised text now clearly states the number of studies for each study design. 9 (173- 176)

"add the references for each study and include the study design of each articles, They study can be arranged according to supply side and demand side issues separately, The table looks mixed of both, and the discussion section also can follow the same." We appreciate this insightful suggestion. We have revised Table 2 to include references for each study and their respective study designs. Furthermore, we have reorganized the table to separate supply-side and demand-side issues.

Regarding the organization of supply-side and demand-side issues, we would like to clarify that our current discussion section follows this structure. The section "Health System Factors" (lines 315-377) addresses supply-side issues, covering topics such as service delivery, health workforce, financing, and governance. The "Contextual Factors" section (lines 379-486) primarily deals with demand-side issues, including cultural beliefs, gender dynamics, and individual-level barriers. We believe this existing structure provides a clear and comprehensive analysis of these distinct but interrelated aspects of diabetes care in West Africa. Pages 10 – 21, Table 2

Reviewer 2

Response

Page, (Lines)

"The format needs to be the same in the whole document. Also, please check all the grammatical errors." We appreciate your attention to detail. We have thoroughly reviewed the manuscript for formatting consistency and grammatical errors. The document has been carefully edited to ensure uniformity in formatting and to correct any grammatical issues.

"Introduction section needs to be concise, it is vast and vague."

Thank you for this valuable feedback. We have thoroughly revised the introduction section, streamlining the content to provide a more focused and targeted overview of the topic. We have eliminated extraneous information and sharpened our focus on the key issues relevant to diabetes care in West Africa. The revised introduction now provides a more concise and clear context for our study.

In effecting these changes, the following references were modified:

Removed:

1. Tesema AG, Ajisegiri WS, Abimbola S, Balane C, Kengne AP, Shiferaw F, et al. How well are non-communicable disease services being integrated into primary health care in Africa: A review of progress against World Health Organization's African regional targets. PLOS ONE. 2020 Oct 22;15(10):e0240984.

2. Galvanizing Action on Primary Health Care: Analyzing Bottlenecks and Strategies to Strengthen Community Health Systems in West and Central Africa | Global Health: Science and Practice [Internet]. [cited 2024 Feb 11]. Available from: https://www.ghspjournal.org/content/9/Supplement_1/S47

3. Koohpayehzadeh J, Azami-Aghdash S, Derakhshani N, Rezapour A, Alaei Kalajahi R, Sajjadi Khasraghi J, et al. Best Practices in Achieving Universal Health Coverage: A Scoping Review. Med J Islam Repub Iran. 2021 Dec 30;35:191.

Added:

Communicable and non-communicable diseases in Africa in 2021/22 | WHO | Regional Office for Africa [Internet]. [cited 2024 Oct 6]. Available from: https://www.afro.who.int/publications/communicable-and-non-communicable-diseases-africa-202122

3, (50 – 105)

"If possible, please give the prevalence."

We appreciate you highlighting the importance of including prevalence statistics. We have added the following information to the introduction:

"As of 2019, NCDs accounted for approximately 37% of all mortality in the region, up from 24% in 2000, with diabetes being a major contributor alongside cardiovascular diseases, cancers, and chronic respiratory diseases (1,2)."

This addition clarifies the increasing impact of NCDs, including diabetes, in the context of the ongoing burden of communicable diseases in West Africa.

3, (52 – 57)

"This needs to be shifted at the end of the introduction section."

As suggested, the research questions have been moved to the end of the introduction section.

This placement provides a smoother transition into the methods section and offers readers an immediate sense of the scope of our review. 5, (97 – 100)

"Add children also"

As suggested, we have corrected the omission of children in the exclusion criteria.

6, Table 1, Row 4

"Add "n" for both of the following" "Please mention "n" for all the study designs."

We have modified the section as follows:

"Twelve papers were included in this scoping review after screening 4,290 documents. Most of the included studies (n=7) were from Nigeria, with the remaining studies conducted in Ghana (n=4) and Senegal (n=1). The study designs employed across the various studies included qualitative (n=7), quantitative (n=4) and a mixed-method study (n=1), with study populations predominantly involving healthcare providers and patients." 9, (168 – 180)

"Please add references along with the name of authors." "Title of the study can be removed from the table. (Title is not relevant here)" "Please add references" We have revised Table 2 to reflect these suggestions. References have been added alongside author names, titles have been removed, and necessary changes have been made.

Pages 10 - 21

We thank you again for your valuable feedback, which has helped to strengthen our manuscript, and look forward to your decision.

Sincerely,

Abdul-Basit Abdul-Samed on behalf of all authors

---

## [Decision Letter · Decision Letter 1]

7 Jan 2025

PGPH-D-24-02021R1

Improving Type II Diabetes Care in West Africa: A Scoping Review of Barriers, Facilitators and the Way Forward

Dear Dr. Abdul-Samed,

Thank you for submitting your manuscript to PLOS Global Public Health. After careful consideration, we feel that it has merit but does not fully meet PLOS Global Public Health’s publication criteria as it currently stands. Therefore, we invite you to submit a revised version of the manuscript that addresses the points raised during the review process.

We look forward to receiving your revised manuscript.

Kind regards,

Ari Probandari, PhD

Academic Editor

Journal Requirements:

Additional Editor Comments (if provided):

Reviewers' comments:

Reviewer's Responses to Questions

**Comments to the Author**

1. If the authors have adequately addressed your comments raised in a previous round of review and you feel that this manuscript is now acceptable for publication, you may indicate that here to bypass the “Comments to the Author” section, enter your conflict of interest statement in the “Confidential to Editor” section, and submit your "Accept" recommendation.

Reviewer #1: All comments have been addressed

Reviewer #3: All comments have been addressed

2. Does this manuscript meet PLOS Global Public Health’s publication criteria ? Is the manuscript technically sound, and do the data support the conclusions? The manuscript must describe methodologically and ethically rigorous research with conclusions that are appropriately drawn based on the data presented.

Reviewer #1: Yes

Reviewer #3: Partly

3. Has the statistical analysis been performed appropriately and rigorously?

Reviewer #1: N/A

Reviewer #3: N/A

4. Have the authors made all data underlying the findings in their manuscript fully available (please refer to the Data Availability Statement at the start of the manuscript PDF file)?

Reviewer #1: Yes

Reviewer #3: Yes

5. Is the manuscript presented in an intelligible fashion and written in standard English?

Reviewer #1: Yes

Reviewer #3: Yes

6. Review Comments to the Author

Reviewer #1: Thanks for the nice study. It became much better than before. In result section, kindly confirm the exclusion criteria added is not repeated in methodology section.

Reviewer #3: Thank you for the opportunity to review this manuscript. The authors highlighted the barriers and facilitators of care to patients with type II diabetes mellitus in the West African sub-region. Find below my few comments:

Title: Consider rephrasing thus: “Barriers and Facilitators of Primary Care Management of Type II Diabetes Mellitus in the West African Sub-Region: A Scoping Review”.

Abstract

Kindly follow the PRISMA-scl checklist as much as possible. For example, in the method sub-section of the abstract, it is required that you state eligibility criteria for the studies included, sources of evidence, and charting methods. Your conclusion should be based on the review questions and objectives.

Background: The background is well-written and provides clarity to the index review. However, the second research question is centered around the mechanism through which these factors work. I am not sure how this was answered in the review.

Methodology:

It may be good to organize this section using the Arksey and O’Malley framework as follows:

Stage 1: Identifying the research question: Here define the research question(s) and establish the scope of the review

Stage 2: Identify relevant studies: Here describe the inclusive search strategy and how the studies were selected (apply the inclusion and exclusion criteria)

stage 3: Study selection: describe the screening process e.g., from title to abstract then the full text

stage 4: Charting the data

stage 5: Collating, summarizing, and reporting the results

These headings should guide the description of the methodology to ensure rigor. The authors already stated that they plan to use the Arksey and colleague framework, it will be good to stick with it. I can see that some of this information is already in the manuscript, kindly insert the headings and expand on your description while focusing on the PRISMA-scl checklist for each heading

Inclusion and Exclusion criteria

• Why the inclusion of reviews and meta-analyses?

• The search strategy described within the text is not exhaustive. Authors should specify the full electronic search strategy for at least one database, including any limits used, such that it could be replicated or repeated.

• There is a contradiction, while in the section for inclusion criteria, the authors included meta-analyses and case reports but in the result section, it was written that non-empirical articles were excluded. What do you mean by non-empirical articles? Are case reports empirical or non-empirical data?

• Kindly highlight the research gaps in the topic before charting the path forward in the discussion section

General Comment: The article is scientifically appropriate, and I recommend acceptance when these comments are addressed or appropriate rebuttal provided.

7. PLOS authors have the option to publish the peer review history of their article (what does this mean? ). If published, this will include your full peer review and any attached files.

---

## [Author Response · Author response to Decision Letter 2]

23 Jan 2025

1/16/2025

Ari Probandari, PhD

Academic Editor

Dear Dr Ari Probandari,

Thank you for considering our manuscript for publication in PLOS Global Public Health, and we appreciate the time and effort of the reviewers whose feedback has been valuable. We have carefully revised the manuscript accordingly, and in the table below includes point-by-point responses and actions taken to address each comment.

Reviewer #1 Comments:

Comment Response Page/Lines

"In result section, kindly confirm the exclusion criteria added is not repeated in methodology section." We have carefully reviewed both the methodology and results sections to ensure consistency in the presentation of exclusion criteria. The criteria are now clearly stated once in the methodology section (Table 1) and referenced appropriately in the results without repetition. Pages 5-9

Reviewer 3 comments:

Comment Response Page (Lines)

Title: Consider rephrasing thus: “Barriers and Facilitators of Primary Care Management of Type II Diabetes Mellitus in the West African Sub-Region: A Scoping Review”.

We thank the reviewer for this suggestion and agree that the proposed title better reflects the core focus of our review questions and findings. We have modified the title accordingly. The new title more precisely captures our systematic analysis of barriers and facilitators in primary care management of diabetes in West Africa, aligning directly with our research questions. N/A

Kindly follow the PRISMA-scl checklist as much as possible. For example, in the method sub-section of the abstract, it is required that you state eligibility criteria for the studies included, sources of evidence, and charting methods. Your conclusion should be based on the review questions and objectives.

We have revised the abstract to strictly follow the PRISMA-ScR checklist. The methods section now clearly states: - Eligibility criteria - Sources of evidence - Charting methods - Search strategy - Study selection process Pages 2-3

Methodology:

It may be good to organize this section using the Arksey and O’Malley framework as follows:

Stage 1: Identifying the research question: Here define the research question(s) and establish the scope of the review

Stage 2: Identify relevant studies: Here describe the inclusive search strategy and how the studies were selected (apply the inclusion and exclusion criteria)

stage 3: Study selection: describe the screening process e.g., from title to abstract then the full text

stage 4: Charting the data

stage 5: Collating, summarizing, and reporting the results

These headings should guide the description of the methodology to ensure rigor. The authors already stated that they plan to use the Arksey and colleague framework, it will be good to stick with it. I can see that some of this information is already in the manuscript, kindly insert the headings and expand on your description while focusing on the PRISMA-scl checklist for each heading Thank you, we take this point and have restructured the methodology section using Arksey and O'Malley's framework, with clear headings for each stage: - Stage 1: Identifying research questions - Stage 2: Identifying relevant studies - Stage 3: Study selection - Stage 4: Charting the data - Stage 5: Collating, summarizing, and reporting results Pages 5-8

Clarify inclusion/exclusion criteria contradictions regarding meta-analyses and case reports We have revised the inclusion/exclusion criteria to remove any contradictions. Secondary research (including meta-analyses) is now clearly listed under exclusion criteria. We have also clarified that only primary research studies were included. The definition of empirical articles has been explicitly stated. Page 7, Table 1

Authors should specify the full electronic search strategy for at least one database, including any limits used, such that it could be replicated or repeated.

We have added Appendix 1 containing the complete search strategy for PubMed, including all search terms, Boolean operators, and limits used, ensuring replicability. Supporting Information (S3)

Kindly highlight the research gaps in the topic before charting the path forward in the discussion section

We have added a new section titled "Research Gaps" before "The Path Forward" in the discussion section, systematically outlining: - Geographic limitations in current evidence - Implementation research gaps - Patient perspective gaps - Digital health intervention evidence gaps - Multisectoral approach evaluation needs Page 36-37

Comment Response Page (Lines)

"The second research question is centered around the mechanism through which these factors work. I am not sure how this was answered in the review." We appreciate this insightful observation. Thank you. We have strengthened the Results section to make the mechanisms more explicit while avoiding redundancy. Specifically, we have:

1) Revised key sections to clearly articulate how different factors operate through distinct mechanisms: - The financial mechanism showing how lack of insurance creates direct barriers to access (p.31) - Service delivery mechanism demonstrating how inadequate facilities and fragmented care affect outcomes (p.31-32) - Cultural mechanism illustrating how beliefs influence care-seeking pathways (p.33) - Psychological mechanism revealing the bidirectional relationship between distress and care outcomes (p.36-37)

2) Added language to show how these mechanisms interact, creating compound effects on diabetes care

3) Maintained the existing structure while making mechanisms more explicit, allowing the Discussion to flow naturally from these established mechanisms and situate them within broader literature These revisions better answer our second research question by clearly demonstrating how factors work to influence diabetes care in West Africa. Pages 31-37

We thank you again for your valuable feedback, which has helped to strengthen our manuscript, and look forward to your decision.

Sincerely,

Abdul-Basit Abdul-Samed on behalf of all authors

---

## [Decision Letter · Decision Letter 2]

12 Mar 2025

Barriers and Facilitators of Primary Care Management of Type II Diabetes Mellitus in the West African Sub-Region: A Scoping Review

PGPH-D-24-02021R2

Dear Dr Abdul-Samed,

We are pleased to inform you that your manuscript 'Barriers and Facilitators of Primary Care Management of Type II Diabetes Mellitus in the West African Sub-Region: A Scoping Review' has been provisionally accepted for publication in PLOS Global Public Health.

Best regards,

Ari Probandari, PhD

Academic Editor

Reviewer Comments (if any, and for reference):

Reviewer's Responses to Questions

**Comments to the Author**

1. If the authors have adequately addressed your comments raised in a previous round of review and you feel that this manuscript is now acceptable for publication, you may indicate that here to bypass the “Comments to the Author” section, enter your conflict of interest statement in the “Confidential to Editor” section, and submit your "Accept" recommendation.

Reviewer #1: All comments have been addressed

Reviewer #3: All comments have been addressed

2. Does this manuscript meet PLOS Global Public Health’s publication criteria ? Is the manuscript technically sound, and do the data support the conclusions? The manuscript must describe methodologically and ethically rigorous research with conclusions that are appropriately drawn based on the data presented.

Reviewer #1: Yes

Reviewer #3: Yes

3. Has the statistical analysis been performed appropriately and rigorously?

Reviewer #1: N/A

Reviewer #3: N/A

4. Have the authors made all data underlying the findings in their manuscript fully available (please refer to the Data Availability Statement at the start of the manuscript PDF file)?

Reviewer #1: Yes

Reviewer #3: Yes

5. Is the manuscript presented in an intelligible fashion and written in standard English?

Reviewer #1: Yes

Reviewer #3: Yes

6. Review Comments to the Author

Reviewer #1: (No Response)

Reviewer #3: I am satisfied with the authors' approach to the comments earlier raised

7. PLOS authors have the option to publish the peer review history of their article (what does this mean? ). If published, this will include your full peer review and any attached files.

**Do you want your identity to be public for this peer review?** For information about this choice, including consent withdrawal, please see our Privacy Policy .

Reviewer #1: **Yes: ** Senthilkumar Ramasamy

Reviewer #3: No
